# Morphological Analysis of Intratesticular Structures Affecting Hamster Testicular Stiffness

**DOI:** 10.3390/ani15202999

**Published:** 2025-10-16

**Authors:** Shiki Hagino, Yoko Sato, Miki Yoshiike, Shiari Nozawa, Kenji Ogawa, Daisuke Tomizuka, Akane Kinebuchi, Yuna Tamakuma, Kohei Ohnishi, Takeshige Otoi, Masayasu Taniguchi, Teruaki Iwamoto

**Affiliations:** 1Department of Animal Reproduction, Joint Faculty of Veterinary Medicine, Yamaguchi University, Yamaguchi 753-0841, Yamaguchi, Japanmasa0810@yamaguchi-u.ac.jp (M.T.); 2Department of Medical Engineering, Faculty of Allied Sciences, University of East Asia, Shimonoseki 751-8503, Yamaguchi, Japan; 3Department of Biology, School of Biological Sciences, Tokai University, Sapporo 005-8601, Hokkaido, Japan; 4Department of Urology, St. Marianna University of Medicine, Kawasaki 216-8511, Kanagawa, Japant4iwa@marianna-u.ac.jp (T.I.); 5Faculty of Science and Technology, Keio University, Yokohama 223-8522, Kanagawa, Japanohnishi@sd.keio.ac.jp (K.O.); 6Bio-Innovation Research Center, Tokushima University, Tokushima 779-3233, Tokushima, Japan; otoi@tokushima-u.ac.jp; 7Division of Male Infertility, Sanno Hospital, Center for Human Reproduction for IVF, International University of Health and Welfare, Minato 107-0052, Tokyo, Japan

**Keywords:** testicular stiffness, spermatogenesis, cryptorchidism, hamster, micro-force sensor, testis morphology, animal model, mechanobiology

## Abstract

**Simple Summary:**

Animal models of testicular dysfunction provide important insights into mechanisms of impaired spermatogenesis. Herein, a hamster model of experimental cryptorchidism was used to investigate the effect of morphological alterations in testicular structures on overall testicular stiffness by using a stiffness measuring robotic system equipped with a micro-force sensor device. Our results suggest that testicular stiffness is related to spermatogenic activity and may serve as a novel biomechanical indicator in animal-based reproductive research.

**Abstract:**

Testicular stiffness is a potential indicator of spermatogenic activity. Herein, we investigated the relationship between testicular stiffness and intratesticular morphology in Syrian hamsters by using a robotic system with a micro-force sensor. Animals were divided into control, sham-operated, and surgically induced cryptorchidism groups. Testicular stiffness, testis weight and size, and Johnsen score data for sham and crypt groups were partially derived from our previous study and reanalysed. Testicular stiffness and histological parameters were analysed, including tunica albuginea thickness, seminiferous tubule occupancy, tubule diameter, intratubular cell-layer thickness, peritubular lamina propria thickness, and Leydig cell numbers. Compared with those of sham and normal controls, cryptorchid testes showed significantly lower stiffness and marked morphological changes, such as reduced tubule occupancy and diameter, thinner intratubular cell layers, thickened tunica albuginea and peritubular lamina propria, and increased numbers of Leydig cells. Decreased testicular stiffness and the Johnsen score, a standard index of spermatogenic function, were strongly related to these structural changes. These findings indicate that structural changes in the testes caused by impaired spermatogenesis are related to measurable differences in tissue stiffness. This study supports using mechanical properties as non-invasive quantitative indices to evaluate testicular function in animal models, offering a novel approach for future research in experimental andrology.

## 1. Introduction

Spermatogenesis, the complex process by which male germ cells are produced, is fundamental to male fertility and reproductive health [1,2]. A key area of interest in reproductive biology is building an understanding of the structural and functional changes associated with impaired spermatogenesis, as such impairments can lead to various forms of male infertility. Consequently, the accurate assessment of testicular structure and function is critical in clinical and research settings.

Animal models provide indispensable platforms for exploring testicular pathophysiology and for developing novel methodologies to assess spermatogenic function. In particular, surgically induced cryptorchidism in animals serves as a well-established model to mimic conditions of spermatogenic arrest [3,4], enabling the morphological and functional investigation of the testes under experimentally controlled settings [5,6,7,8].

In recent years, efforts have been made to quantify testicular properties beyond conventional histology [3,4,9,10,11]. Biomechanical characteristics of testicular tissue, such as hardness, may reflect underlying changes in spermatogenic status and cellular architecture. Previously, we introduced a novel method for quantifying testicular hardness using a robotic system equipped with a micro-force sensor, demonstrating its potential in evaluating spermatogenesis in a hamster model of experimental cryptorchidism [12]. Our findings demonstrated a significant reduction in testicular stiffness in cryptorchid testes, with positive correlations to the Johnsen score (JS) [13], which is a histological index of spermatogenesis. This study highlighted the potential of mechanical measurements as functional markers of testicular health in animals.

In our previous studies, we highlighted correlations between the thickening of the lamina propria of seminiferous tubules and impaired spermatogenesis in human testes [14], alongside alterations in the composition of the extracellular matrix [15]. A change in the composition of the extracellular matrix would likely impact the physical properties of the walls of the seminiferous tubules. Therefore, it is conceivable that the hardness of the seminiferous tubules and that of the testes may be altered in testes with impaired spermatogenesis.

Building on this concept, the present study aimed to analyse in greater detail how structural features within the testis contribute to overall testicular hardness by employing a comprehensive approach in a controlled Syrian hamster model and integrating precise stiffness measurements with detailed histomorphometric analysis. By examining parameters such as tunica albuginea thickness, seminiferous tubule occupancy, tubule diameter, intratubular cell-layer thickness, peritubular lamina propria thickness, and number of Leydig cells, we sought to uncover the structural basis of altered testicular biomechanics in the context of impaired spermatogenesis [10,11,16].

## 2. Materials and Methods

### 2.1. Animals

All animal and sample collection procedures were reviewed and approved by the Animal Ethics Committee and the Institutional Review Board of the University of St. Marianna (#1601013, #1612009, #1712012, and #1902011; approved on 2 February 2016, 30 January 2017, 6 February 2018, and 3 February 2019, respectively). Mature male Syrian hamsters aged 11–14 weeks were housed in the laboratory animal facility at St. Marianna University School of Medicine (light/dark cycle: 12 h:12 h; 22 °C, 50–55% humidity). To ensure acclimatisation to the housing environment, the hamsters were pre-reared for 1 week before the commencement of the experiment.

An experimental animal model of spermatogenesis failure due to cryptorchidism was created. Cryptorchid lesions in the testis indicate abnormal spermatogenesis. The left testis was removed intraperitoneally and fixed under isoflurane inhalation anaesthesia (FUJIFILM Wako Pure Chemical Co., Tokyo, Japan). Subsequently, both testes were orchiectomised after a 2-week interval to measure and characterize their pathologies and morphologies. This study comprised a cryptorchidism (crypt group) (n = 16) and a sham examination (sham group; n = 15), with an additional untreated group designated as the normal (normal group; n = 12). Furthermore, the epididymides from each group of samples were removed and used for the evaluation of spermatogenesis.

The crypt and sham group data for testicular weight, size, stiffness, and JS [13] were obtained from our previously published dataset [12]. In the present study, we added a normal group and measured the same parameters to serve as a baseline control. To further investigate the relationship between testicular stiffness and internal structure, we performed detailed image-based morphometric analysis on all groups. The following parameters were included: tunica albuginea thickness, seminiferous tubule occupancy (as percentage of testicular cross-sectional area), seminiferous tubule diameter, intratubular cell-layer thickness, seminiferous tubule peritubular lamina propria thickness, and numbers of Leydig cells.

### 2.2. Measurements of Testicular Weight, Size, and Stiffness

After washing the surface of the removed testis with saline, excess moisture was wiped off with Kimwipes (Nippon Paper Crecia Co., Ltd., Tokyo, Japan). Subsequently, the testicular weight and size were measured on a Falcon Culture dish (Corning, Tokyo, Japan). The testicular weight was measured using a weighing scale, and the testicular size was determined by measuring the long and short diameters. The 5-point stiffness of the centre of the testis, 3 mm to the right and left and 1 mm up and down from the centre, was measured using our previously developed robotic device tool equipped with a micro-force sensor to measure the stiffness of testicular tissue [12]. To prevent desiccation of the testes during measurement, they were maintained in a moistened saline-filled wet box.

### 2.3. Tissue Processing

To preserve the morphology and composition of the tissue, the testes and epididymides were immediately fixed in Bouin’s solution (FUJIFILM Wako Pure Chemical Co., Osaka, Japan) for 24 h. Following fixation, the tissues were washed three times for 1 h in phosphate-buffered saline (FUJIFILM Wako Pure Chemical Co., Osaka, Japan), followed by dehydration with serial alcohol concentrations (70–100%) for 1 h each, and then replaced with xylene (FUJIFILM Wako Pure Chemical Co., Osaka, Japan) three times for 1 h. Subsequently, the tissue samples were replaced with xylene dissolved in paraffin wax (Sigma-Aldrich Japan., Tokyo, Japan) (1:1) and were impregnated in paraffin wax three times for 1 h before routine embedding. Serial sections of 4 μm thickness were then obtained using a rotary microtome and mounted on slides. The morphology and developmental stages of the testes within each group were assessed through haematoxylin (Muto Pure Chemicals, Co., Ltd., Tokyo, Japan) and eosin (Muto Pure Chemicals, Co., Ltd.,Tokyo, Japan) staining, with sections containing tumours excluded from the analysis post-staining.

### 2.4. Evaluation of JS

The JS is a 10-point index used to quantitatively evaluate the spermatogenic potential of human testes, with a higher score indicating a greater spermatogenesis ability [13]. To assess spermatogenesis in hamsters, the human JS was adapted with certain modifications [12]; specifically, scores of 5 and below were assessed using human criteria. Briefly, each score showed the presence of cell types in seminiferous tubules: score 1: no cells; score 2: no germ cells but Sertoli cells were present; score 3: spermatogonia were the only germ cells present; score 4: only few spermatocytes (<5) and no spermatids or spermatozoa were present; score 5: no spermatozoa and no spermatids but several or many spermatocytes were present. Spermatozoa maturation stages, including round spermatids, initial elongated spermatids, and late elongated spermatids, were assigned scores 6, 6.5, and 7, respectively, representing the highest scores attainable. Furthermore, thirty randomly selected seminiferous tubules from each sample were evaluated to determine the average JS.

### 2.5. Observation and Morphological Evaluation

All images were captured using a (Nikon Ecript NiU-TRFLM; Nikon, Tokyo, Japan) microscope equipped with a Nikon DS-Fi2-U3 digital camera (Nikon, Tokyo, Japan), connected to a computer. The NIS-Elements D 4.10.00 (Nikon, Tokyo, Japan) software was used for the morphological evaluation of testicular tissues and epididymal tissues. The examined fields were randomly selected by examiners who were blinded to the tissue groups. Each testicular sample was analysed for parameters, including seminiferous tubule occupancy rate, testicular tunica albuginea thickness, seminiferous tubule diameter, cell-layer thickness of seminiferous tubule, peritubular lamina propria thickness, and numbers of Leydig cells. Each epididymal tissue was checked to note whether it contained sperm or not.

### 2.6. Measurement of Thickness of Testicular Tunica Albuginea

The tunica albuginea is a dense fibrous connective tissue composed mainly of collagen fibres present on the surface of the testis (Figure 1A). In instances where the tunica albuginea was partially detached or lost during sectioning, areas adjacent to the testicular parenchyma were selected and used for the measurements. The thickness of the testicular tunica albuginea was measured at 10 locations per sample, and the average was calculated.

### 2.7. Measurement of Seminiferous Tubule Occupancy Rate in Testicular Tissues

Because the diameter tube of the seminiferous tubule is cylindrical, both circular and elliptical cross-sections of the diameter tube may coexist in the same sample. To ensure consistency, a field of view was selected where the cross-sections of the seminiferous tubule were more densely circular. The entire area of this selected field of view was measured, and the total area of the seminiferous tubules was measured by manually circling the seminiferous tubules individually in the image (Figure 1B). The seminiferous tubule occupancy percentage (%) was calculated using the results of each measurement were used to obtain (area of seminiferous tubules/total area) × 100 in one field of view.

### 2.8. Measurement of Diameter of Seminiferous Tubules

Initially, a seminiferous tubule with a more circular cross-section was selected from the sample. Subsequently, the diameter (length and width) of each seminiferous tubule was measured, and the average for each tubule was calculated (Figure 1C). This process was repeated for 10 seminiferous tubules per sample and the resulting measurements were averaged to derive a representative value.

### 2.9. Measurement of Cell-Layer Thickness in the Seminiferous Tubules

The seminiferous tubules contain both spermatogenic cells and Sertoli cells, with HE-stained areas defining the cell layer within the seminiferous tubules. To determine the diameter of the seminiferous tubules, the thickness of the cell layer was measured in each of the 10 selected tubules. (Figure 1D). Five cell layers were randomly selected and measured per seminiferous tubule, resulting in a total of 50 measurements per sample, and the average was calculated.

### 2.10. Measurement of the Thickness of the Peritubular Lamina Propria of the Seminiferous Tubules

The peritubular lamina propria is situated on the basal surface of the seminiferous tubule epithelium (Figure 1E). The peritubular lamina propria is the area comprising the basement membrane, myoid cells, and the acellular layers. The thickness of the peritubular lamina propria of the seminiferous tubules was measured in each of the 10 seminiferous tubules selected for diameter assessment. Ten peritubular lamina propria per seminiferous tubule were randomly selected and measured. A total of 100 measurements were taken per sample, and the average was calculated.

### 2.11. Measurement of the Numbers of Leydig Cells

The numbers of Leydig cells were visually counted in restricted fields of view per group of histological sections. Data were recorded as the mean number of Leydig cells in the interstitial area (situated on the basal surface of the seminiferous tubule epithelium) of five fields per sample.

### 2.12. Azan Staining and Immunostaining

All procedures were performed at 25 °C, unless otherwise specified. To detect collagen fibres, Azan staining [17] was used with slight modifications. The sections were de-paraffinised in xylene, then rehydrated using a descending graded ethanol solution, and washed with distilled water. The sections were incubated for 1 h with preheated Mallory azocarmine G (Muto Pure Chemicals, Co., Ltd., Tokyo, Japan) at 50 °C. Incubation was then continued for an additional 1h at 25 °C. After washing with tap-water, the sections were differentiated with aniline alcohol solution (Muto Pure Chemicals, Co., Ltd., Tokyo, Japan). After this, the sections were washed with acetic acid alcohol and then with running tap water. Thereafter, the slides were incubated with phosphotungstic acid solution (Muto Pure Chemicals, Co., Ltd.,Tokyo, Japan) for 2 h and washed with distilled water. Finally, the sections were incubated with Mallory’s aniline blue-orange G mixture (Muto Pure Chemicals, Co., Ltd., Tokyo, Japan) for 30 min and differentiated with 100% ethanol. After immersion in xylene, the sections were sealed with Eukit quick-hardening mounting medium (Fluka Analytical, Tokyo, Japan) and embedded with a coverslip.

To detect myoid cells, alpha smooth muscle actin (αSMA) antibody immunohistochemistry was performed. After deparaffinisation and rehydration, the sections were washed with PBS and treated with 10 mM citric acid buffer (pH 6.0) (Sigma-Aldrich, Tokyo, Japan) for antigen retrieval in a microwave for 30 min. To prevent endogenous peroxidase activity, the sections were blocked with 0.3% hydrogen peroxide (H_2_O_2_) solution (FUJIFILM Wako Pure Chemical Co., Osaka, Japan) in methanol (FUJIFILM Wako Pure Chemical Co., Osaka, Japan) for 1 h. After preincubation of the sections with normal goat immunoglobulin G (IgG, 500 μg/mL; Dako, Tokyo, Japan) dissolved in 1% bovine serum albumin (Sigma-Aldrich, Tokyo, Japan) in PBS for 1 h to block non-specific bindings, primary mouse monoclonal antibody against αSMA (Nichirei Co., Tokyo, Japan) was applied to the sections and incubated overnight. The negative control was a matched mouse IgG isotype control (Dako, Tokyo, Japan) at the working concentration of the primary antibody. The sections were then incubated with goat anti-mouse IgG (Fab’) and labelled with an amino acid polymer–peroxidase complex (Histofine Simple Stain MAX PO (M), Nichirei Co., Tokyo, Japan) for 30 min. Sections were visualised using 3-amino-9-ethylcarbazol (DAB) (Histofine Simple Stain DAB solution, Nichirei Co., Tokyo, Japan) for 20 min. The nuclei in the sections were counterstained with haematoxylin solution and mounted using Eukit quick-hardening mounting medium (Fluka Analytical, Tokyo, Japan) with a glass coverslip after rehydration in xylene.

### 2.13. Statistical Analysis

The statistical program JMP ver.12 (SAS Institute Inc., Tokyo, Japan) was used for statistical analysis. Analysis of variance (ANOVA) and Tukey’s multiple comparison tests were performed to investigate differences among crypt, sham, and normal groups. The relationships between each testicular parameter, JS, and stiffness value were analysed using the Kruskal–Wallis test. Statistical significance was determined based on probability values (*p* ≤ 0.05), with differences meeting this criterion considered statistically significant.

## 3. Results

### 3.1. Testicular Weight

Testicular weight was significantly lower in the crypt group (0.61 ± 0.12 mg) than in the sham group (1.86 ± 0.15 mg) and normal group (1.83 ± 0.23 mg) (*p* < 0.05) (Table 1). No significant differences were observed between the sham and normal groups. These findings suggest that neither abdominal surgery nor anaesthesia had a significant effect on testicular weight, while 2 weeks of cryptorchid administration resulted in a decrease in testicular weight.

### 3.2. Testicular Size

In both the long vertical and short horizontal axes of testicular size, the crypt group (15.2 ± 1.22 mm, 10.4 ± 1.15 mm) was significantly lower than the sham group (20.9 ± 0.80 mm, 14.73 ± 0.70 mm) and the normal group (20.92 ± 0.70 mm, 14.58 ± 0.70 mm) (*p* < 0.05) (Table 1). Conversely, no significant differences were observed between the sham and normal groups. Therefore, neither abdominal surgery nor anaesthesia influenced testicular size, while the testes experienced a reduction in size following 2 weeks of cryptorchidism.

### 3.3. Testicular Stiffness Value

The testicular stiffness values of the crypt group (0.40 ± 0.13 μN/μm) were predominantly lower than those of the sham group (1.46 ± 0.31 μN/μm) and the normal group (1.39 ± 0.25 μN/μm) (*p* < 0.05). No differences were observed between the sham and normal groups. Neither abdominal surgery nor anaesthesia had any effect on testicular stiffness, while the testicular stiffness decreased after 2 weeks of cryptorchism.

### 3.4. JS

The average JS in the crypt group was 4.08 ± 0.71 (Table 1), indicating a scarcity of spermatocytes with a notable absence of sperm and spermatids (Figure 2A). However, the JS of the crypt group varied widely between samples, with some testes exhibiting a JS of 3 (Figure 2B) or exceeding 5 (Figure 2C,D), indicative of partial spermatogenesis (Figure 2E). The average JS values of the sham group and normal group were 6.96 ± 0.04 and 6.94 ± 0.05, respectively (Table 1); both groups showed a large number of sperm and spermatids, many of which were late elongated spermatids. The JS of the crypt group was significantly lower than that of the other two groups (*p* < 0.05) (Table 1). Since no difference was observed between the sham and normal groups, it suggests that neither abdominal surgery nor anaesthesia had any effect on spermatogenesis, and JS decreased after 2 weeks of cryptorchidism.

### 3.5. Sperm in Epididymides

All epididymides in the normal groups and sham groups were filled with plenty of sperm. However, most epididymides with cryptorchid testes did not contain sperm. In rare cases where small numbers of seminiferous tubules containing round spermatids or spermatids were present in the cryptorchid testes, little sperm was observed.

### 3.6. Relationship Between JS and Stiffness Values

Higher JS indicating better spermatogenic function in the sham and normal groups was associated with higher stiffness values; however, lower JS indicating lower spermatogenic function in the crypt group was associated with lower stiffness value (Figure 3). The crypt group had completely different values of both stiffness and JS (*p* < 0.0001). This suggests that the testes with better spermatogenic function tended to exhibit greater stiffness.

### 3.7. Thickness of Testicular Tunica Albuginea

The thickness of the testicular tunica albuginea was significantly higher in the crypt group (82.59 ± 24.73 µm) compared with both the sham group (25.91 ± 12.81 µm) and the normal group (30.67 ± 10.82 µm) (*p* < 0.05) (Table 1 and Figure 4A). Conversely, no significant disparity was observed between the sham and normal groups, suggesting that neither abdominal surgery nor anaesthesia influenced the thickness of the testicular tunica albuginea, and it was found that the testes thickened after 2 weeks of cryptorchidism.

Higher JS and higher stiffness values in the sham and normal groups were associated with lower values of testicular tunica albuginea thickness; however, lower JS and lower stiffness values in the crypt group were associated with higher values of testicular tunica albuginea thickness. This implies that, as the thickness of the testicular tunica albuginea increased, both spermatogenic function and testicular stiffness values decreased (Figure 5A and Figure 6A).

### 3.8. Seminiferous Tubule Occupancy Rate in Testicular Tissues

Seminiferous tubule occupancy was significantly lower in the crypt group (82.57 ± 7.55%) than in the sham group (93.32 ± 2.23%) and the normal group (93.69 ± 1.77%) (*p* < 0.05) (Table 1 and Figure 4B). Conversely, no differences were observed between the sham and normal groups. Therefore, both abdominal surgery and anaesthesia did not affect the occupancy rate of the seminiferous tubules. However, a decrease in JS was noted after 2 weeks of cryptorchidism.

Furthermore, higher JS and higher stiffness values in the sham group and normal group were associated with a higher seminiferous tubule occupancy rate; however, lower JS and lower stiffness values in the crypt group were associated with a lower seminiferous tubule occupancy rate. This suggests that, as seminiferous tubule occupancy increased, both spermatogenic function and testicular stiffness values also increased (Figure 5B and Figure 6B).

### 3.9. Seminiferous Tubule Diameter

Seminiferous tubule diameter was significantly lower in the crypt group (165.67 ± 19.94 µm) than in the sham group (267.49 ± 14.19 µm) and the normal group (264.67 ± 17.89 µm) (*p* < 0.05) (Table 1 and Figure 4C). There was no difference between the sham and normal groups, indicating that neither abdominal surgery nor anaesthesia had any effect on the diameter of seminiferous tubules, and it was revealed that the diameter decreased after 2 weeks of cryptorchidism.

Higher JS and higher stiffness values in the sham group and normal group were associated with higher seminiferous tubule diameter values; however, lower JS and lower stiffness values in the crypt group were associated with lower seminiferous tubule diameter value. As the diameter of the seminiferous tubules increased, both spermatogenic function and the testicular stiffness value increased (Figure 5C and Figure 6C).

### 3.10. Thickness of Intraductal Cell Layers of the Seminiferous Tubule

There was a significant difference in the cell-layer thickness of the seminiferous tubule in the crypt group (34.59 ± 9.61 µm), the sham group (75.57 ± 10.25 µm), and the normal group (84.37 ± 5.40 µm), all of which displayed significant differences (*p* < 0.05) (Figure 4D). While the effects of abdominal surgery and anaesthesia on the cell layer in the seminiferous tubules were observed, the thickness in the crypt group was significantly lower than those of the other two groups (*p* < 0.05), indicating that the thickness of the cell layer in the seminiferous tubule was reduced after 2 weeks of cryptorchidism.

In addition, higher JS and higher stiffness values in the sham group and normal group were associated with higher cell-layer thickness values; however, lower JS and lower stiffness values in the crypt group were associated with lower cell-layer thickness values. This indicates that, as the thickness of the cell layer in the seminiferous tubule increased, both spermatogenic function and testicular stiffness values also increased (Figure 5D and Figure 6D).

### 3.11. Peritubular Lamina Propria Thickness of Seminiferous Tubule

The peritubular lamina propria thickness of the seminiferous tubule was significantly thicker in the crypt group (1.67 ± 0.16 µm) than in the sham group (1.14 ± 0.08 µm) and the normal group (1.06 ± 0.06 µm) (*p* < 0.05) (Table 1, Figure 4E). Conversely no significant difference was observed between the sham and normal groups, suggesting that neither abdominal surgery nor anaesthesia had any effect on the peritubular lamina propria of the seminiferous tubules. This indicates that the peritubular lamina propria of the seminiferous tubules thickened after 2 weeks of cryptorchidism.

Furthermore, higher JS and higher stiffness values in the sham and normal groups were associated with lower values of peritubular lamina propria thickness; however, lower JS and lower stiffness values in the crypt group were associated with higher thickness of peritubular lamina propria values. This indicates that, as the thickness of the peritubular lamina propria increased, both spermatogenic function and the testicular stiffness values decreased (Figure 5E and Figure 6E).

### 3.12. Numbers of Leydig Cells

Numbers of Leydig cells were significantly higher in the crypt group (85.54 ± 17.81) than in the sham group (23.76 ± 4.14) and the normal group (25.46 ± 6.33) (Table 1, Figure 4F). Conversely no significant difference was observed between the sham and normal groups, suggesting that neither abdominal surgery nor anaesthesia had any effect on the numbers of Leydig cells. This indicates that the numbers of Leydig cells increased after 2 weeks of cryptorchidism.

In addition, higher JS and higher stiffness values in the sham group and normal group were associated with lower numbers of Leydig cells; lower JS and lower stiffness values in the crypt group were associated with higher numbers of Leydig cells. This indicates that, as the numbers of Leydig cells increased, both spermatogenic function and the testicular stiffness values decreased (Figure 5F and Figure 6F).

## 4. Discussion

This study investigated the relationship between testicular stiffness and intratesticular structural parameters in a Syrian hamster model of experimentally induced cryptorchidism. By combining micro-force sensing technology with detailed histomorphometric analysis, we demonstrated that changes in testicular stiffness were closely related to histological alterations associated with impaired spermatogenesis. These findings support the concept that tissue biomechanics, specifically stiffness, may serve as a quantitative indicator of testicular health and function.

Our results showed that testicular stiffness was significantly reduced in the cryptorchid group than in both the sham and normal groups. This finding extends with previous findings of Ogawa et al. [12], who reported reduced stiffness in cryptorchid hamster testes, with concurrent decreases in spermatogenic activity. Importantly, the inclusion of a normal (untreated) control group in our study strengthens the interpretation that the reduction in stiffness observed in the cryptorchid group was not merely due to surgical manipulation but rather related to pathophysiological changes associated with cryptorchidism itself. Furthermore, our study extended this observation by quantitatively analysing specific morphological features—such as tunica albuginea thickness, seminiferous tubule occupancy, seminiferous tubule diameter, cell-layer thickness, peritubular lamina propria thickness, and numbers of Leydig cells. These changes occurred alongside alterations in the stiffness of the testicular tissue and alongside alterations in the JS, reinforcing the link between microstructural organization and spermatogenic status. In the present study, we could analyse only a certain time point (2 weeks post-cryptorchidism); in the near future, we aim to obtain more information with the analysis of various stages of the experimental cryptorchidism process and the use of a model for a seasonal study to compare the state of stiffness during periods of rest and sexual activity, because the hamster breeds during seasons with long days and is at rest during seasons with short days [18].

Changes in the hardness of biological tissues have been a focus of attention in many medical fields. While assessing tissue hardness is crucial, it poses challenges in conditions like cancerous tissue and live cirrhosis compared with normal tissues [19,20,21]. There are many methods for measuring hardness; however, ultrasound is the most widely used method in the medical field [22,23,24,25]. Nevertheless, the use of ultrasound has certain limitations; the hardness value changes depending on the radiation range of the ultrasonic waves, making it unsuitable for the measurement of very small specific areas or small testes. Therefore, we examined the usefulness of a measurement device using a stiffness value, which is an index of hardness, for the quantitative determination of spermatogenesis capacity.

In the case of testes for measurement stiffness, there are several reports of humans with cryptorchid and varicocele, although they do not include information on testis morphology [26,27]. For animals, elastographic and morphological testicular changes in hypothyroidism are reported; however, treated experimental rat testes were also reported to show spermatogenesis [28]. Interestingly, the present findings are in line with results from canine models. Gloria et al. showed that testicular stiffness, as estimated by strain elastography, reflected spermatogenic activity and histological integrity in dogs [29]. In their study, testicular stiffness negatively correlated with the tubule area, diameter, and epithelial thickness but not with the connective tissue content. These parallels between species suggest that changes in seminiferous tubule structure, rather than interstitial fibrosis, primarily account for alterations in testicular stiffness. It should be noted, however, that the degree of spermatogenic impairment differs substantially between their canine model and our cryptorchid hamster model. In Gloria et al.’s study, the spermatogenic capacity was evaluated in dogs that were still producing sperm, with assessments based on sperm number, motility, and morphology from epididymal sperm [29]. In contrast, our model represents a more severe form of spermatogenic dysfunction in which spermatogenesis is mostly arrested owing to experimentally induced cryptorchidism. Our data similarly indicate that reduced tubule occupancy and cell-layer thickness contribute to decreased stiffness in cryptorchid testes, supporting the idea that seminiferous tubule degeneration is a key biomechanical determinant.

The use of a robotic micro-force sensing system in this study allowed for the direct high-resolution measurement of tissue stiffness, offering advantages over elastographic methods that rely on image interpretation [30,31]. The assessment of severe fibrosis/cirrhosis in the liver by strain elastography shows lower performance than measurement by transient elastography [25]. Moreover, by reanalysing stiffness and JS data from our previous work in combination with newly acquired histological measurements, we were able to strengthen the mechanistic understanding of decreased testicular hardness in the context of spermatogenic failure.

Based on the present results, we elucidated factors affecting testicular hardness (Figure 7) from cross-sections of testes and seminiferous tubules from the normal (sham) and crypt groups. This discussion focused on both the entire testis and the seminiferous tubules. When examining the reduction in stiffness values across the entire testis (Figure 7A,B), the thickness of the testicular tunica albuginea of the crypt group was increased, and, if it was uniformly thickened under the same contents, the stiffness value should increase. However, the observed decrease in stiffness values in the crypt group presented conflicting results. One possible explanation is that when the tunica albuginea thickened, some components were replaced by softer tissue, resulting in overall thickening but decreased stiffness. Indeed, the tunica albuginea of the crypt group had lighter staining tones than those of the sham and normal groups, and some parts of the albuginea had three-layered structures (Figure 8A,B). Collagen fibre accumulation in the tunica albuginea has been reported in rats after the administration of drugs to induce the reduction in spermatogenesis [32]. However, in a rat model, the tunica albuginea thickness did not change after drug administration. In our study, the tunica albuginea in both groups contained collagen fibres; however, the density differed (Figure 8C,D). Furthermore, we need to conduct further investigation about the type of collagen fibre to clarify their contents. Another potential factor could be the decrease in seminiferous tubule occupancy rate in the crypt group. This implies an increase in the interstitial area, and it is possible that the interstitium also had a disproportionate increase in soft tissue components. However, future analysis of individual tissue components is necessary to confirm this hypothesis. Additionally, the decrease in testicular size observed in the crypt group suggests a corresponding reduction in internal pressure within the testes, which could contribute to the overall decrease in stiffness.

Next, we considered the decrease in the stiffness value of the seminiferous tubules (Figure 7C,D). In the crypt group, the diameter of the seminiferous tubules was smaller, the cell layers within the tubules were thinner, and the peritubular lamina propria was thickened. In the crypt group, collagen fibres of the peritubular lamina propria appeared to increase with peritubular lamina propria (Figure 9A,B). However, the changes in the thickness of the peritubular lamina propria were less pronounced compared to changes in the thickness of the cell layer and the diameter of the seminiferous tubules. Furthermore, despite this thickening of the peritubular lamina propria, the combined thickness of the entire tubule (cell layer thickness + peritubular lamina propria thickness) was lower in the crypt group compared with the other two groups. This suggests that, in the crypt group, the overall thickness of the seminiferous tubule was reduced, which may have reduced the stiffness values within the crypt group. Since the testis is a collection of seminiferous tubules, a collective decrease in stiffness values across these tubules likely resulted in the overall decreased stiffness values in the crypt group. In rodents, unlike in humans [33] and boars [34], thickened peritubular layers of lamina propria and several layers of myoid cells do not appear in the testis with cryptorchid or deteriorated spermatogenesis. Hamster testes showed a single layer of myoid cells, similar to that in other rodents; however, the αSMA expression appeared to be increased in cryptorchid testes (Figure 9C,D), similar to that in bovine testicular seminiferous tubules showing deteriorated spermatogenesis [35].

Furthermore, the present study showed that the seminiferous occupancy rate was significantly lower in the cryptorchid group than in the normal and sham group (Figure 4B), suggesting that the interstitium area was wider in the crypt group. In addition, the number of Leydig cells increased in the interstitium in the crypt group (Figure 4F). This result is consistent with reports from other animals showing deteriorated spermatogenesis [36]. There is a possibility that the stiffness may be affected by the numbers of Leydig cells; however, further analysis is needed on the molecular characteristics of Leydig cells.

Recent strides in reproductive medicine have opened avenues for previously untreatable male infertility, notably through techniques such as intracytoplasmic sperm injections [37]. Microscopic intravital testicular sperm retrieval (MD-TESE) has gained attention recently and is now used as a treatment for patients with nonobstructive azoospermia, a condition previously considered an absolute barrier to fertility [38,39]. MD-TESE involves the extraction of sperm from seminiferous tubules in the testis, boasting a success rate of approximately 40–60% [40,41,42]. Improving the sperm recovery rate is a challenge; an index for selecting seminiferous tubules containing good spermatogenic cells is needed to increase the sperm recovery rate [43]. In the testes of infertile individuals, mosaicism occurs during spermatogenesis, making it difficult to identify specific regions where spermatogenesis proceeds normally and sperm are present [42]. If it becomes feasible to assess the hardness of seminiferous tubules quantitatively, it will be possible to identify the seminiferous tubules harbouring sperm without resorting to extensive tissue sampling from the testes. In the present study, we did not directly measure the stiffness value of individual seminiferous tubules. If the device used in this study could be improved to measure the stiffness of individual seminiferous tubules, we would be able to better understand the relationship between testicular stiffness and spermatogenesis failure. Thus, an improved device that can measure the hardness of the seminiferous tubules could provide useful information for MD-TESE in human.

However, some limitations should be acknowledged. First, the cryptorchid model, while useful, represents a relatively acute and artificial form of spermatogenic disruption. Long-term or more gradual models (e.g., aging [44,45] and hormonal suppression [46]) also influence morphological changes in testicular tissue and may yield different biomechanical signatures. Second, although we analysed multiple structural parameters, other contributors to tissue stiffness—such as extracellular matrix composition or the molecular biological properties of interstitial cell populations—were not directly assessed. Finally, our findings are based on a rodent model, and further validation across species and disease states is warranted, because structural changes in the tubule walls of testes showing deteriorated spermatogenesis vary among the animals.

## 5. Conclusions

This study demonstrated that testicular stiffness, as measured by a robotic micro-force sensing system, related to structural changes in the testis associated with impaired spermatogenesis in a cryptorchid hamster model. Specifically, reductions in seminiferous tubule occupancy, tubule diameter, and intratubular cell-layer thickness, along with thickening of the tunica albuginea, peritubular lamina propria, and increased numbers of Leydig cells, were closely associated with decreased testicular stiffness and lower JSs. These findings support the validity of testicular stiffness as a quantitative biomechanical marker of spermatogenic function. Further research should assess its utility across different models and species, as well as under various factors (such as different stages of the experimental cryptorchidism process, testis under different photoperiod environments, etc).

## Figures and Tables

**Figure 1 animals-15-02999-f001:**
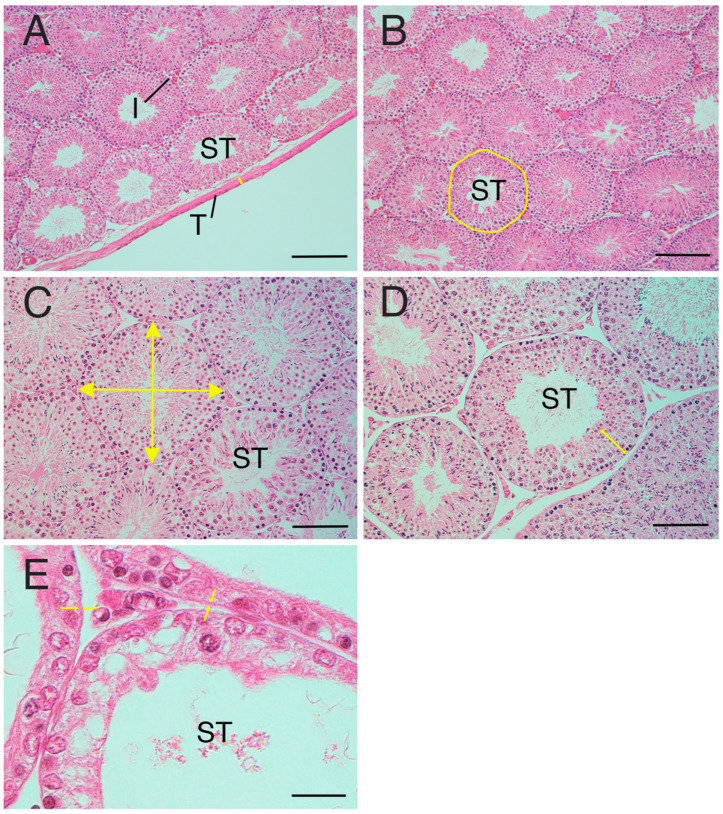
Measurement factors of testicular tissue. (**A**) Tunica albuginea thickness (arrows). (**B**) Seminiferous tubule occupancy rate. Area with line: manually selected seminiferous tubule. (**C**) Diameter of seminiferous tubule (arrow). (**D**) Cell-layer thickness (arrows) in the seminiferous tubule. (**E**) Peritubular lamina propria thickness. Arrows: edge of the peritubular lamina propria. I—interstitial area; T—tunica albuginea; ST—seminiferous tubule. Bar: 200 µm (**A**,**B**); 100 µm (**C**,**D**); 50 µm (**E**).

**Figure 2 animals-15-02999-f002:**
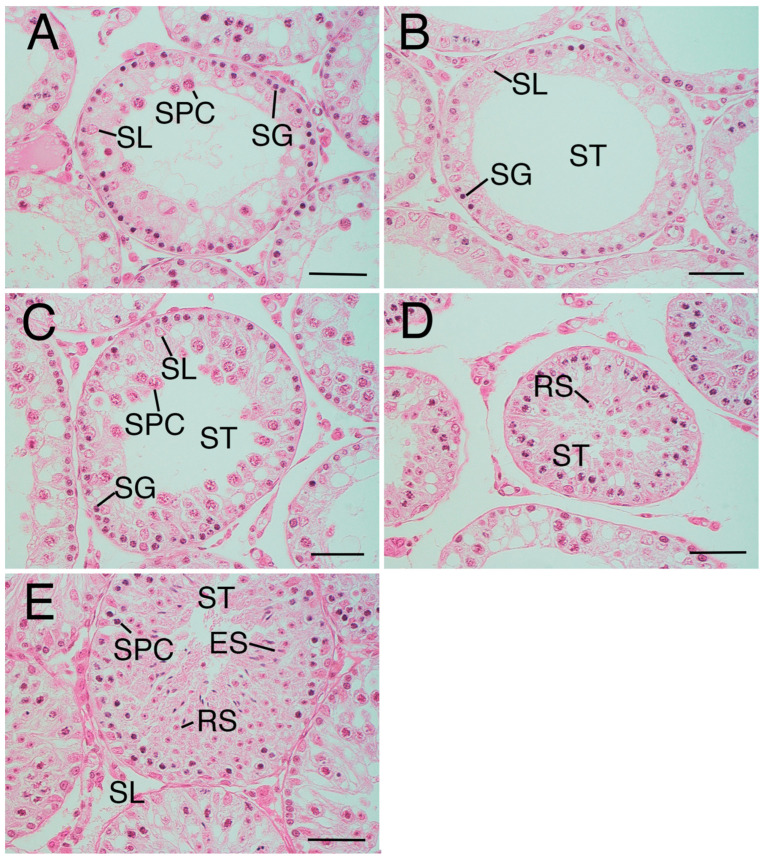
Variation of Johnsen score (JS) in the crypt group. (**A**) Seminiferous tubule in crypt group (mean Johnsen score (JS): 3.96). The tubule only includes spermatogonia (SG), Sertoli cells (SL), and a few spermatocytes (SPC), not spermatid or sperm. (**B**) Seminiferous tubule in crypt group (mean JS:2.93). The tubule only includes SG and SL, not SPC, spermatid, or sperm. (**C**) Seminiferous tubule in the crypt group (mean JS:5.43). (**D**) Same tissue sample as (**C**). Round spermatid (RS) was also observed. The tubule only includes SG, SL, and many SPC, not elongated spermatid or sperm. (**E**) Seminiferous tubule in crypt group (mean JS:6.97). Complete spermatogenesis. ST—seminiferous tubule. Bar: 50 µm.

**Figure 3 animals-15-02999-f003:**
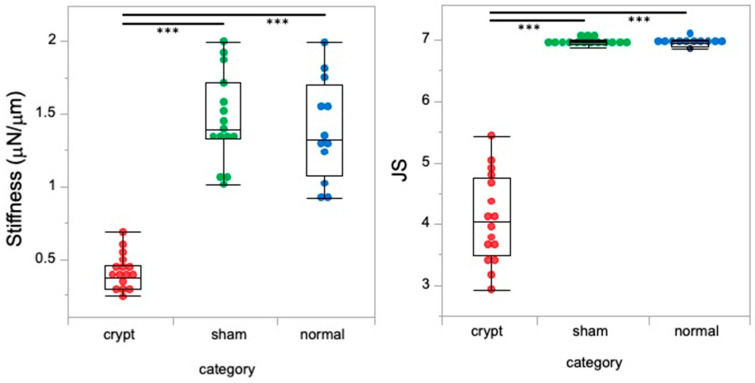
Relationship between JS and stiffness values. Red: crypt group, green: sham group, blue: normal group, ***: *p* < 0.0001.

**Figure 4 animals-15-02999-f004:**
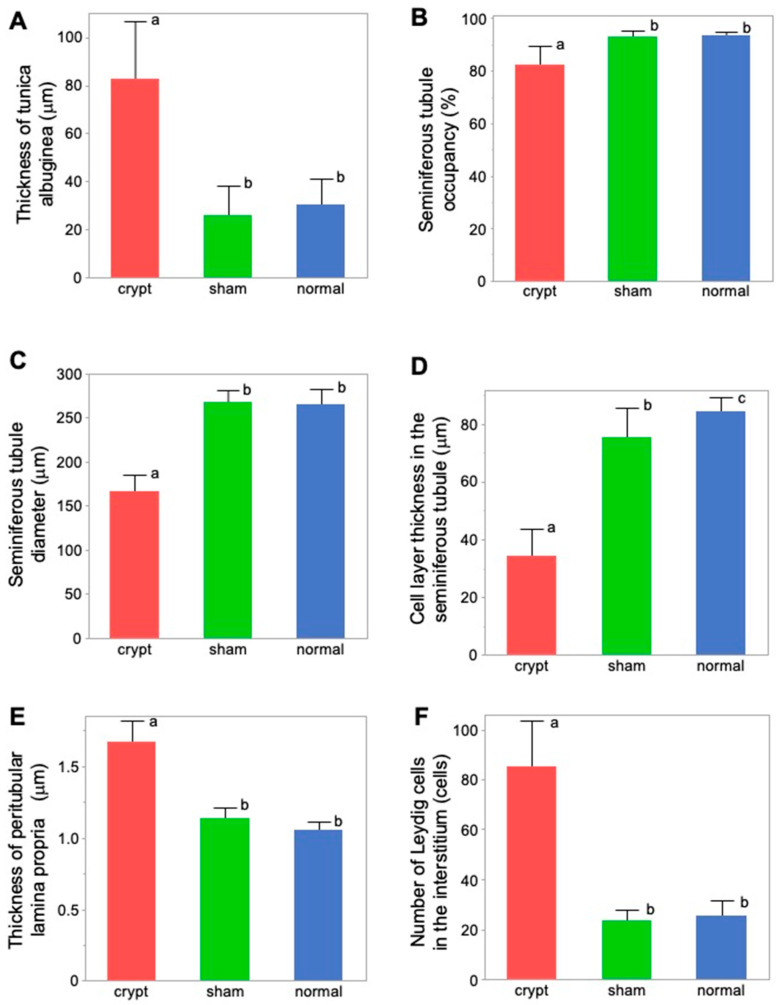
Comparison of the measurement factors of testicular tissues among 3 groups. (**A**) Tunica albuginea thickness. (**B**) Seminiferous tubule occupancy rate. (**C**) Seminiferous tubule diameter. (**D**) Cell-layer thickness in the seminiferous tubule. (**E**) Thickness of peritubular lamina propria. (**F**) Number of Leydig cells. Different letters represent significant differences among different signs (*p* < 0.05).

**Figure 5 animals-15-02999-f005:**
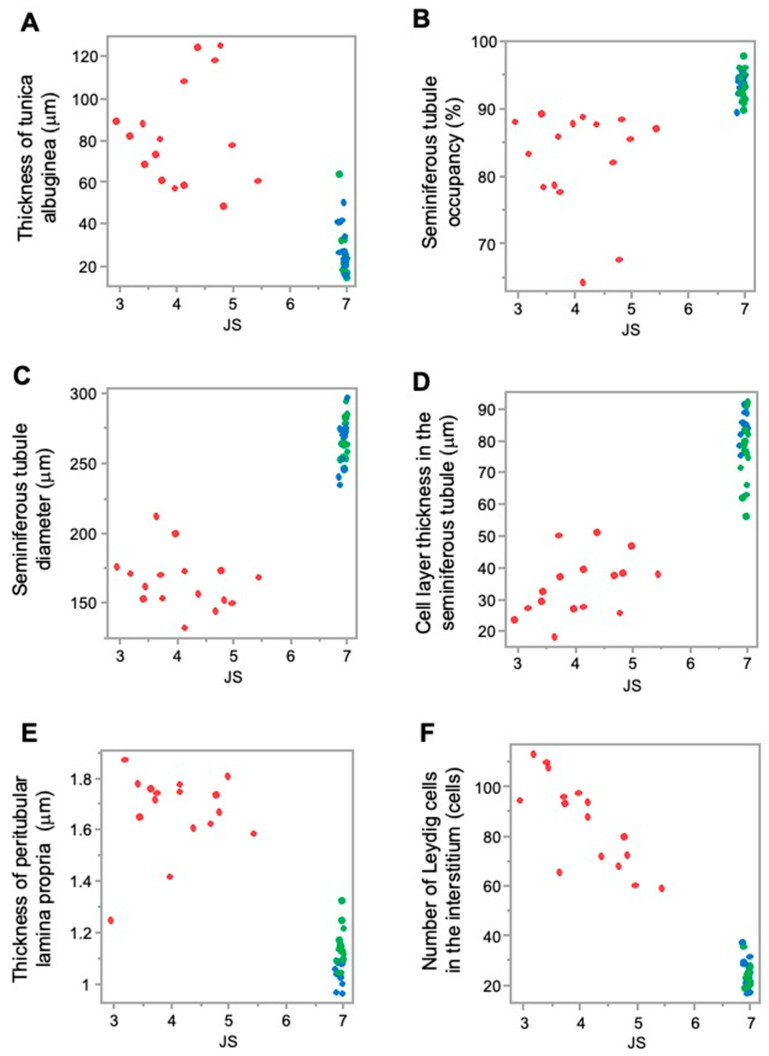
Relationship between the measurement factors of testicular tissues and JS. (**A**) Tunica albuginea thickness; (**B**) seminiferous tubule occupancy rate; (**C**) seminiferous tubule diameter; (**D**) cell-layer thickness in the seminiferous tubule; (**E**) thickness of peritubular lamina propria; and (**F**) number of Leydig cells. Red: crypt group, green: sham group, blue: normal group.

**Figure 6 animals-15-02999-f006:**
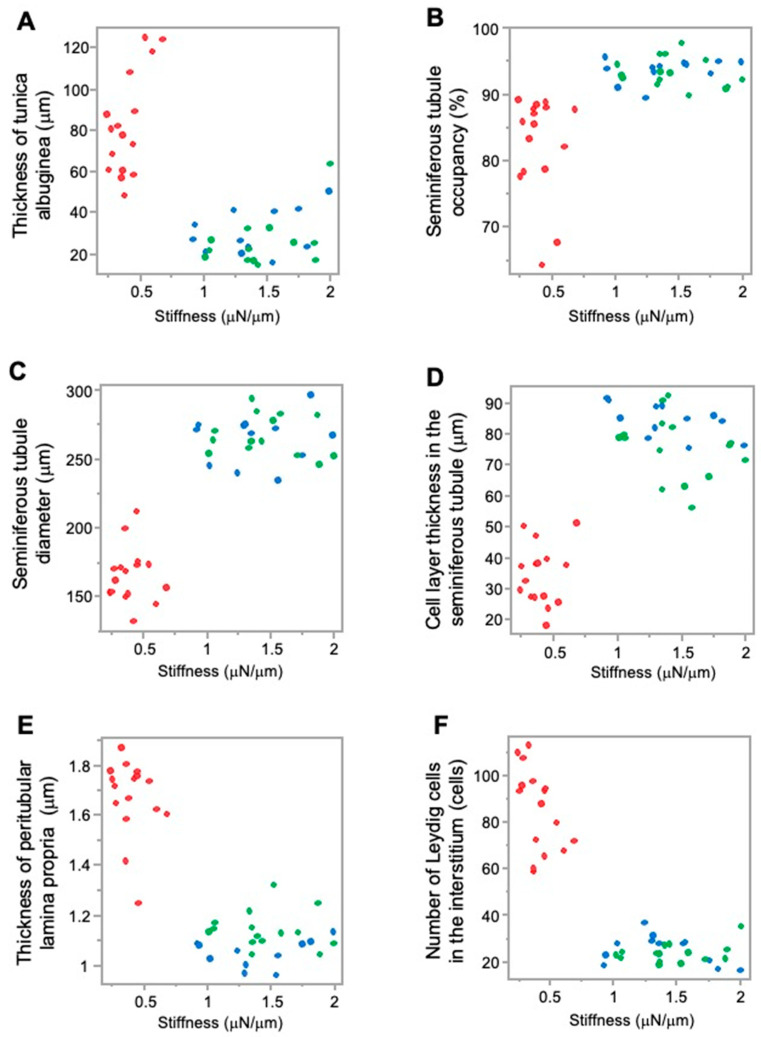
Relationship between the measurement factors of testicular tissues and stiffness. (**A**) Tunica albuginea thickness; (**B**) seminiferous tubule occupancy rate; (**C**) seminiferous tubule diameter; (**D**) cell-layer thickness in the seminiferous tubule; (**E**) thickness of peritubular lamina propria; (**F**) number of Leydig cells. Red: crypt group, green: sham group, blue: normal group.

**Figure 7 animals-15-02999-f007:**
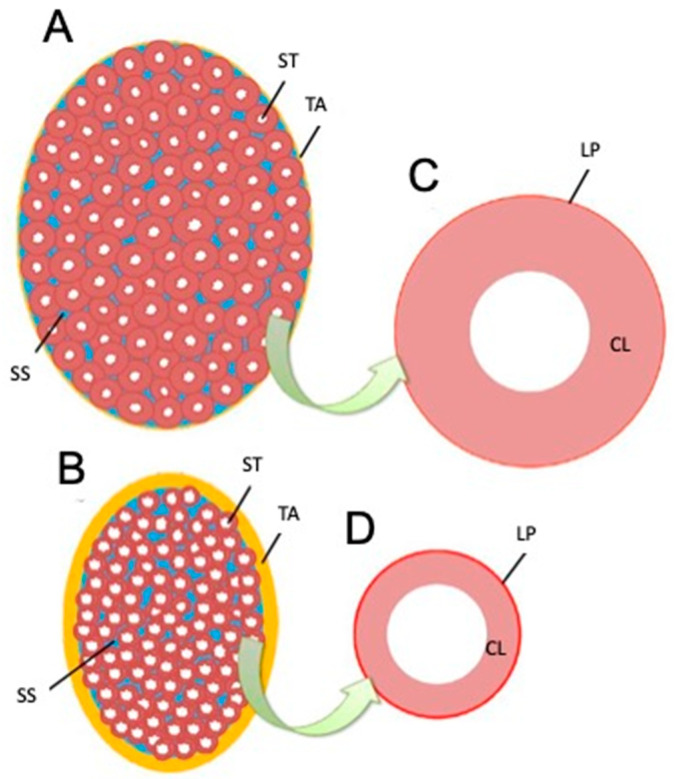
Schematic drawing of testis and seminiferous tubule cross-sections based on measurement results. (**A)** Longitudinal section of the testis in an animal showing normal spermatogenesis. (**B**) Longitudinal sections of testes in animals showing abnormalities. (**C**) Section of seminiferous tubules in (**A**). (**D**) Section of seminiferous tubules in (**B**). ST—seminiferous tubule, SS—interstitial (blue), TA—tunica albuginea (yellow), LP—peritubular lamina propria (red), CL—cell layer in the seminiferous tubule.

**Figure 8 animals-15-02999-f008:**
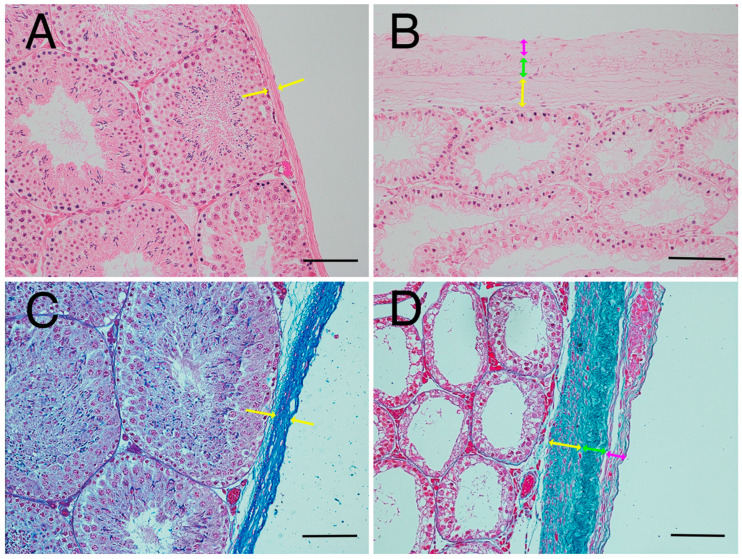
Thickened tunica albuginea. (**A**) Tunica albuginea in the normal group. Thickness of tunica albuginea is indicated between the arrowheads. (**B**) Tunica albuginea in the crypt group. (**C**) Azan-stained tunica albuginea in the normal group. (**D**) Azan-stained tunica albuginea in the crypt group. Thickened tunica albuginea layers with 3 different collagen fibre densities (yellow arrows, green arrows, pink arrows) were observed. Bar: 100 µm.

**Figure 9 animals-15-02999-f009:**
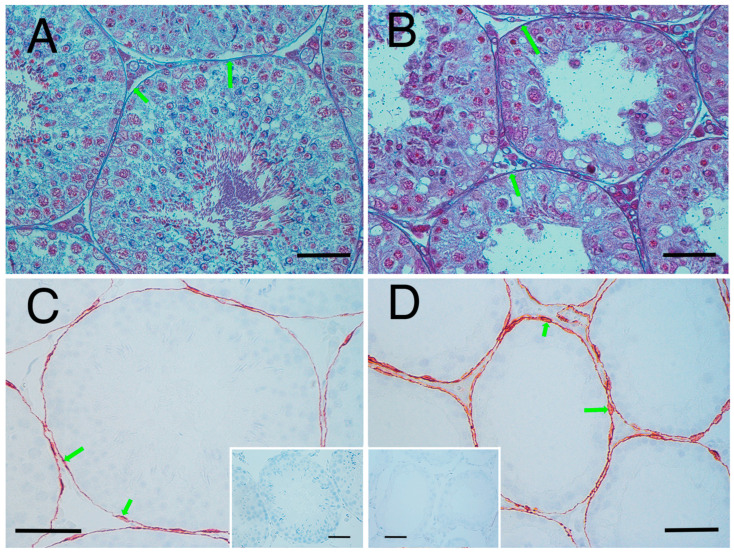
Seminiferous tubule wall. Azan stain: (**A**) normal group and (**B**) crypt group. Arrowheads indicate the appearance of collagen fibres. αSMA immunostaining; (**C**) normal group and (**D**) crypt group. Insets: negative control. Arrowheads show the myoid cells. Bar: 50 µm.

**Table 1 animals-15-02999-t001:** Comparison of various parameters among the three groups.

Parameter	Crypt Operation Group (n = 16)		Sham Operation Group (n = 15)		Normal Group (n = 12)	
	mean ± SD	Range	Mean ± SD	Range	Mean ± SD	Range
Weight (mg)	0.61 ± 0.12 ^a^	0.42–0.89	1.86 ± 0.15 ^b^	1.52–2.08	1.83 ± 0.23 ^b^	1.15–2.05
Testicular size:						
Long diameter (mm)	15.2 ± 1.22 ^a^	13–18	20.9 ± 0.80 ^b^	19–22	20.92 ± 0.70 ^b^	20–22
Short diameter (mm)	10.4 ± 1.15 ^a^	8–13	14.7 ± 0.70 ^b^	14–16	14.58 ± 0.70 ^b^	14–16
Stiffness (µN/µm)	0.40 ± 0.13 ^a^	0.24–0.68	1.46 ± 0.31 ^b^	1.01–2.00	1.39 ± 0.25 ^b^	0.92–1.99
Johnsen score	4.08 ± 0.71 ^a^	2.93–5.43	6.96 ± 0.04 ^b^	6.9–7.0	6.94 ± 0.05 ^b^	6.85–7.0
Thickness of tunica albuginea (µm)	82.59 ± 24.73 ^a^	48.65–125.15	25.91 ± 12.81 ^b^	14.87–63.94	30.67 ± 10.82 ^b^	15.98–50.48
Seminiferous tubule occupancy (%)	82.57 ± 89.24 ^a^	64.30–89.24	93.32 ± 2.23 ^b^	89.85–97.80	93.69 ± 1.77 ^b^	89.50–95.64
Seminiferous tubule diameter (µm)	165.67 ± 19.94 ^a^	132.59–212.27	267.49 ± 14.19 ^b^	246.48–294.33	267.49 ± 14.19 ^b^	234.66–296.99
Thickness of intraductal cell layer (µm)	34.59 ± 9.61 ^a^	18.41–51.35	75.57 ± 10.25 ^b^	56.31–92.32	84.37 ± 5.40 ^c^	75.49–91.51
Thickness of peritubular lamina propria (µm)	1.67 ± 0.16 ^a^	1.25–1.87	1.14 ± 0.08 ^b^	1.05–1.32	1.06 ± 0.06 ^b^	0.96–1.15
Numbers of Leydig cells	85.54 ± 17.81 ^a^	60.2–113	23.76 ± 4.14 ^b^	18.8–27.6	25.46 ± 6.33 ^b^	16.4–37

Different letters represent significant difference among different signs (*p* < 0.05).

## Data Availability

Data supporting the findings of this study are available from the corresponding author upon reasonable request.

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
