# Peer review of "Morphological Analysis of Intratesticular Structures Affecting Hamster Testicular Stiffness"

_animals, 2025, doi:10.3390/ani15202999_

Round 1
Reviewer 1 Report
Comments and Suggestions for Authors
This is a very good manuscript which describe the morphological changes of seminiferous tubules in hamsters designed to examine the effects of surgically induced cryptorchidism. The results is interesting and carefully observed and analysed. However, two issues must be clarified before publishing.
- Although the Johnsen score is a popular method to analysis the development of seminiferous tubules in literature, it is still needed to give some brief explains for hamster in this manuscript, especially some criteria should be included.
- This manuscript is focused on seminiferous tubules, however, the character of testicular Leydig cells, for example relative area percentage, is better to include.
- I am very want to know whether there are mature sperm in the epididymides of Crypt group or not.
Reviewer 2 Report
Comments and Suggestions for Authors
The work was conducted using a simple methodology tailored to the study's objective. There are also certain questions that I believe should be addressed before the work can be accepted for publication:
- a) One limitation is that measurements were not taken at various stages of the experimental cryptorchidism process. The results would better reflect the parallelism between the tissue changes assessed and the decrease in stiffness.
- b) Rodents present a lesion of the tubular wall that typically does not result in tubule sclerosis. This last change the seminiferous tubule into a solid structure. The thickening of the peritubular lamina propria is less in rodents with seminiferous tubule lesions, probably because they present a single layer of myoid cells, unlike what occurs in humans or boars , where tubular sclerosis occurs. This fact could be mentioned in the discussion when discussing the limitation of the results due to the species studied, given that perhaps the tubules lose stiffness due to the disappearance of the seminiferous epithelium. They increase some stiffness in their tubular wall, but not enough to compensate for the loss.
- b) It would have been interesting to assess stiffness with respect to total measurements: interstitium or tubular volume, and tunica albuginea volume. Furthermore, performing a collagen staining technique would have allowed for a correlation between collagen volume in these compartments and stiffness.
- c) The quality of the images does not meet the standards for publication. They lack adequate contrast, a clear background, and focus.
- d) The concept of basement membrane is not well used in the text. The authors refer to the increase in the thickness of the tubular wall (peritubular lamina propria). In the conclusion, I would not use the word "reflect" unless it is correlated or associated.
- In material and methods, it would be appropriate to briefly describe how the cryptorchid lesion in the testis occurred.
Reviewer 3 Report
Comments and Suggestions for Authors
Testicular function is a complex process influenced by several factors, including sex steroids, androgen-dependent proteins, prostaglandins, the extracellular matrix, Sertoli cells, and the effects of aging... The objective of this study is to investigate the relationship between testicular rigidity and intratesticular structural parameters in hamsters with experimental cryptorchidism by combining microforce sensing technology with detailed histomorphometric analysis.
The author showed that changes in testicular rigidity reflect histological alteration linked to impaired sperm production using new biomechanical morphometric indicators (Johnsen scores) in the study of animal reproduction.
-The alteration of spermatogenesis in the testis can be better understood using these methods, which can also help determine the causes of male infertility.
- The author presented various results ranging from histological parameters and their correlation with testicular rigidity and spermatogenesis using various statistical tests to justify the most significant results.
-The few bibliographic references are recent, and only one comparison is with the canine species. Have no other studies been conducted on rigidity related to cryptorchidism?
-The author did not show whether these new methods deserve to be used in humans to highlight the causes of male infertility related to Cryptorchidism.
-The hamster is a long days breeder and is at rest during short days. It would be interesting to use it as a model for a seasonal study to compare the state of rigidity during periods of rest and sexual activity.
- I do not have English language skills, so I can not judge this aspect.
-The author respected the guidelines for authors.
Conclusion: The author presented very good, clear, precise, and significant results. I recommend acceptance of this article given its originality and the diversity of new techniques used in testicular morphology.
